# Transcriptomic Analysis of Salicylic Acid Promoting Seed Germination of Melon under Salt Stress

Miao Yan [1,†], Jiancai Mao [1,†], Ting Wu [1], Tao Xiong [1], Quansheng Huang [2], Haibo Wu [1] and Guozhi Hu [1,*]

1 Hami Melon Research Center, Xinjiang Academy of Agricultural Sciences, Urumqi 830091, China; yanmiao0901@126.com (M.Y.)
2 Research Institute of Nuclear Technology and Biotechnology, Xinjiang Academy of Agricultural Sciences, Urumqi 830091, China
* Correspondence: huguozhi@xaas.ac.cn
† These authors contributed equally to this work.

**Abstract:** This study investigated how salicylic acid (SA) mediates the response of melon (*Cucumis melo*) seeds to salt stress using physiological and transcriptomic methods. The effects of SA on the antioxidant enzymes, osmoregulatory substances, and transcriptome of melon seeds under salt stress were investigated using sodium chloride (NaCl, 100 mmol·L$^{-1}$) as the stress stimulant and SA + NaCl (0.25 mmol·L$^{-1}$ + 100 mmol·L$^{-1}$) as the alleviation treatment. The results showed that SA positively influences salt tolerance by increasing the activity of superoxide dismutase activity (SOD) and catalase activity (CAT) while decreasing proline content (Pro). Differentially expressed genes (DEGs) were identified by transcriptome data analysis, of which 2958 were up-regulated, and 2157 were down-regulated. These genes were mainly involved in the mitogen-activated protein kinase (MAPK) signaling pathway and plant hormone signal transduction, lipid metabolism (linoleic and α-linolenic fatty acid metabolism), biosynthesis of secondary metabolites (phenylpropanoid pathway and flavonoid biosynthesis), and related pathways. Further analysis revealed that SA might alleviate salt stress by initiating a series of signaling pathways under salt stress, participating in lignin biosynthesis to improve cell wall stability, and positively regulating lipoxygenase (LOX) genes. These results provide valuable information and new strategies for future salt resistance cultivation and high melon yield.

**Keywords:** *Cucumis melo*; salicylic acid; salt stress; transcriptome; differential genes

## 1. Introduction

Soil salinity is a serious constraint to sustainable agricultural development, which impedes agricultural production and threatens agricultural profits and global food security. Approximately 20% of arable land worldwide is adversely affected by soil salinity [1,2]. The high concentration of sodium (Na$^+$) ions in saline soils induce osmotic stress and ion toxicity, leading to dysfunctional plant metabolism, as well as oxidative stress and impaired photosynthesis [3–6]. Furthermore, it significantly inhibits seed germination, lateral root formation and biomass and can even lead to plant death [7]. Therefore, improving the salt resistance of crop plants has attracted widespread attention worldwide.

Seed germination is the first and most critical stage of plant morphogenesis, growth and development, as well as a key link that determines the quality of seedling growth [8]. Unsuitable environmental conditions such as salt stress can readily compromise seed germination rate, leading to weak seedling growth and yield reduction [9]. Therefore, studying the germination of melon seeds under salt stress is highly important for production. Previous studies have shown that the accumulation of reactive oxygen species (ROS) in seed species severely inhibits seed germination [10,11]. Additionally, superoxide dismutase (SOD), peroxidase (POD), and catalase (CAT) are essential enzymes involved

in ROS metabolism, and an increase in their activity contributes to the scavenging of reactive oxygen species [12]. Lipoxygenase (LOX) is closely related to plant physiological and biochemical processes such as plant seeds and resistance to adversity stress. It has been found that the plant LOX metabolic pathway can produce substances such as hydroperoxides and oxygen radicals, which have an important role in plant resistance to environmental stress [13].

Salicylic acid (SA) is a natural phytohormone signaling molecule that has an important role in regulating plant responses to biotic and abiotic stresses [14,15]. Physiological mechanisms of SA to alleviate salt stress include the induction of an antioxidant defense system and alleviation of membrane lipid peroxidation [16], reduction of ion toxicity [17], and regulation of cross-talk between hormones or signaling substances [18]. However, hardly any studies have reported that SA mediated the molecular response to salt stress on the germination of melon seeds. With the rapid development of molecular biology and high-throughput sequencing technology, plant adversity research has entered the era of transcriptomics. Using transcriptome sequencing (RNA-seq) technology, researchers can comprehensively and dynamically detect plants' overall gene expression changes during different developmental stages and conditions [19].

Melon is a widely cultivated economic crop with a sweet taste and high nutritional value [20,21]. Yet, soil salinity and other adversities often affect seed germination and the growing period, constituting a prominent problem in melon production. In the present study, we noted that SA promoted melon seed germination under salt stress. The possible mechanism of action was further explored through integrated physiological and transcriptomic analyses to further elucidate the role of SA in regulating seed germination and salt tolerance. These results provide valuable information and new strategies for future salt resistance cultivation and high melon yield.

## 2. Materials and Methods

### 2.1. Plant Materials and Germination Treatments

The melon variety "Nasmi" was provided by the Hami Melon Research Center of Xinjiang Academy of Agricultural Sciences, China.

The full, uniform-sized seeds were disinfected with 1% sodium hypochlorite, washed 3–5 times with distilled water, placed in a 9 cm sterile Petri dish containing two layers of filter paper, and germinated under dark conditions in a constant temperature incubator at 28 °C. Based on the pre-test, the following four treatments were set up: distilled water (CK), 0.25 mmol·L$^{-1}$ salicylic acid (SA), 100 mmol·L$^{-1}$ sodium chloride (NaCl) and NaCl + SA (100 mmol·L$^{-1}$ + 0.25 mmol·L$^{-1}$) for seed germination test. Seeds were considered to germinate when the radicle length exceeded 2 mm, and 90 seeds were evenly placed in each treatment with four replications. The germination potential was calculated on the 2nd day of germination, and the germination rate was calculated on the 4th day of germination with the following equations [22]:

Germination rate (%) = (number of germinated seeds by day 4/total number of test seeds) × 100.

Germination potential (%) = (number of germinated seeds by day 2/total number of test seeds) × 100.

At 4 days of germination, 30 seedlings were randomly selected from each treatment. The full length of the shoot (from the bottom of the radicle to the tip of the embryo axis) was measured with a straightedge, the surface water was blotted out with absorbent paper, and the fresh mass was weighed with an electronic balance.

### 2.2. Determination of SOD, CAT, POD Activities, and Proline (Pro) Content

A total of 1 g (approximately 100 seeds) of seeds with the seed coat removed were collected at 0 h, 6 h, 48 h, 60 h and 72 h, and physiological indicators such as antioxidant enzymes were determined.

SOD (U/g FW) activity was measured with a superoxide dismutase activity test kit (Norminkoda Biotechnology Co., Ltd., Wuhan, China). First, crude enzyme extract was

prepared. We weighed 0.1 g tissue, added 1 mL 0.5 mol/L phosphate buffer (pH 7.8), and mixed it via $10,000 \times g$ 4 °C centrifugation for 10 min. The supernatant was the crude enzyme solution. Then, 45 μL of 100 μmol/L EDTA-Na2 solution, 100 μL of 750 μmol/L nitroblue tetrazolium solution, 3 μL of xanthine oxidase, 18 μL of the sample, and 35 μL of 130 mmol/L methionine solution were added to 96-well plates. The control tube included 18 μL of double-distilled water instead of the sample. After mixing, the samples were incubated for 30 min at room temperature, and absorbance was read at 560 nm. SOD enzyme activity was calculated based on fresh weight. At a percentage inhibition in the above xanthine oxidase conjugate reaction system of 50%, SOD enzyme activity in the reaction system was defined as unit enzyme activity [23].

CAT (U/g FW) was measured with a catalase activity test kit (Norminkoda Biotechnology Co., Ltd., Wuhan, China). $H_2O_2$ has a characteristic absorption peak at 240 nm. Catalase can decompose $H_2O_2$, making the absorbance of the reaction solution at 240 nm decrease with reaction time. The activity of catalase can be characterized according to the change rate of absorbance [24].

POD (U/g FW) was measured with a peroxidase activity test kit (Norminkoda Biotechnology Co., Ltd., Wuhan, China). First, 50 mmol/L acetate buffer (pH 5.5), 0.5 mol/L $H_2O_2$ solution, and 25 mol/L guaiacol solution were placed at 25 °C for more than 10 min. For the test, 120 μL of 50 mmol/L acetate buffer, 30 μL of 0.5 mol/L $H_2O_2$ solution, 30 μL of 25 mol/L guaiacol solution, 60 μL of distilled water, and 5 μL of the sample were added successively into EP tubes and mixed. Then, 200 μL of the reaction was immediately transferred to 96-well plates, and absorbance at 30 s was read at 470 nm (A1). Absorbance A2 was obtained after 1 min 30 s with the activity defined as POD activity per gram of fresh tissue. A change of 0.005 in POD was defined as unit enzyme activity [25]. Pro (μg/g FW) content was determined by the biochemical kits (Norminkoda Biotechnology Co., Ltd., Wuhan, China) [26].

### 2.3. RNA-Seq Library Preparation and Sequencing

Three replicates of each NaCl and SA + NaCl treatment were collected at 48 h and stored at −80 °C after a quick freeze in liquid nitrogen for transcriptome sequencing and reverse transcription-quantitative PCR (qRT-PCR) analysis. The transcriptome sequencing was performed by Metware Biotechnology Co., Ltd. (Wuhan, China). The eukaryotic mRNA was enriched by magnetic beads with Oligo (dT) beads; the mRNA was broken into short fragments by adding an interruption reagent. Then, the single-strand cDNA was synthesized with six-base random primers using it as a template; the double-stranded cDNA was purified, and the polymerase chain reaction (PCR) was amplified. Raw data obtained from the sequencing platform were filtered to remove reads with adapter and N ratios > 10% to remove low-quality reads (the number of bases with quality $Q \leq 10$ accounted for more than 50% of the entire reads) and to obtain high-quality clean data for subsequent information analysis.

### 2.4. De Novo Assembly, Functional Annotation, and Enrichment Analysis of Differentially Expressed Genes

The melon genome database (http://cucurbitgenomics.org/organism/18, accessed on 20 May 2021) was used as the reference sequence for the annotation of high-quality clean reads. The threshold value of $|\log_2 \text{Fold Change}| \geq 1$ and $p$-value $\leq 0.05$ was used. The false discovery rate (FDR) was obtained by correcting for the $p$-value. The accepted Benjamini–Hochberg correction method was used to correct the $p$-values for the original hypothesis test, and the FDR was finally adopted as the key indicator for differentially expressed gene screening. The UniGene sequences in the KEGG, NR, Swiss-Prot, GO, and KOG databases were annotated for gene function using DIAMOND BLASTX software (San Francisco, CA, USA). The expression levels of genes were estimated using the FPKM (fragments per kilobase of transcript per million mapped reads) formula.

### 2.5. qRT-PCR Analysis

To verify the accuracy and reliability of transcriptome sequencing, 12 significantly differentially expressed genes were randomly selected and analyzed using qRT-PCR. RNA was extracted using a TIANGEN RNAsimple (Tiangen Biotech, Beijing, China) kit [27], and cDNA was synthesized using PrimeScript TM1ST Strand cDNA Synthesis Kit (TaKaRa, Kusatsu, Japan). The primer sequences used for the qRT-PCRanalysis were designed by Primer ExpressV3.0 (Table S1). The *CmADP* gene was used as an internal reference [28], and the ABI7500 quantitative PCR instrument was used for real-time fluorescence quantitative PCR. Each sample was repeated three times. The relative expression was calculated according to the $2'^{-\Delta\Delta Ct}$ method [29].

### 2.6. Statistical Analysis

Statistical analysis was carried out with SPSS 22.0 software (Chicago, IL, USA) by using one-way analysis of variance (ANOVA), followed by Tukey's test. Differences were considered significant at $p \leq 0.05$. Data are presented here as means $\pm$ SD from at least three measurements. Heatmaps of DEGs were generated using the TBTools software (version 1.068, San Francisco, CA, USA).

## 3. Results

### 3.1. Effect of Different Treatments on Germination of Melon Seeds

As shown in (Table 1), salt stress significantly reduced the germination index and seedling quality of melon seeds. Compared with the CK treatment, the germination rate and germination potential under salt stress decreased by 43.34% and 51.78%, respectively. SA significantly increased the germination rate and germination potential of melon seeds under salt stress, as well as the fresh weight and seedling length. On the other hand, SA application alone increased the germination potential of seeds; however, germination rate, fresh weight, and seedling length were all slightly lower compared to the control treatment but with non-significant differences.

**Table 1.** Effect of different treatments on germination of melon seeds.

| Treatment | Germination Potential/% | Germination Rate/% | Fresh Weight/g | Seedling Length/cm |
|---|---|---|---|---|
| CK | 91.75 ± 0.04 ab | 99.75 ± 0.00 a | 4.39 ± 0.08 a | 8.91 ± 0.29 a |
| SA | 97.00 ± 0.00 a | 99.25 ± 0.04 a | 4.35 ± 0.05 a | 8.31 ± 0.29 a |
| SA + NaCl | 89.50 ± 0.01 b | 93.75 ± 0.01 a | 3.80 ± 0.16 b | 6.75 ± 0.37 b |
| NaCl | 44.25 ± 0.02 c | 56.50 ± 0.03 b | 1.55 ± 0.03 c | 0.96 ± 0.17 c |

Note: Different lowercase letters indicate significant differences at the 0.05 probability level ($p < 0.05$) according to and Tukey's multiple range tests.

### 3.2. Salicylic Acid (SA)-Mediated Physiological Response to Salt Stress

As shown in Figure 1A–C, salt stress decreased the POD, SOD, and CAT activities of melon seeds at 6–48 h of germination compared to CK. SA treatment increased SOD and CAT activities of melon seeds under salt stress throughout the treatment period, and both were the first to show significant differences at 48 h, with a 1.23- and 1.03-fold increase compared to NaCl. In this experiment, only POD activity was found to be significantly increased at 48 h and 72 h. SA application alone significantly increased SOD and CAT activities at 60 h. Under salt stress, the overall trend of Pro content was "rising-declining-rising", while SA reduced Pro content under salt stress throughout the treatment time, reaching the maximum difference value at 48 h with a significant reduction of 30.72%. No significant difference between SA and CK was observed (Figure 1D).

From 0 to 6 h, no seeds of melon germinated under four treatments; yet, from 48 to 60 h, SA obviously promoted the germination of melon seeds under salt stress, and the difference appeared first at 48 h (Figure S1). Combining the phenotypes and the above physiological indicators, 48 h was finally selected as the sampling point for transcriptome samples.

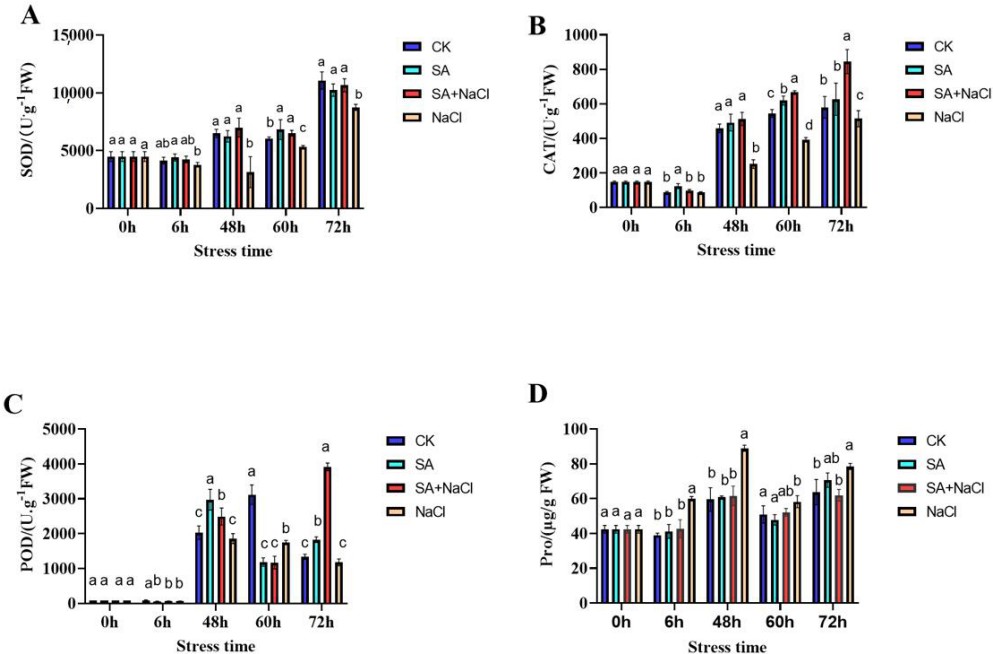

**Figure 1.** Effect of SA on physiological indicators of melon seeds under salt stress. (**A**) Superoxide dismutase activity (SOD); (**B**) catalase activity (CAT); (**C**) peroxidase activity (POD); (**D**) proline content (Pro). Different lowercase letters indicate significant differences at the 0.05 probability level ($p < 0.05$) according to and Tukey's multiple range tests.

### 3.3. Transcriptome Data Analysis

Three replicates of two treatments treated for 48 h were analyzed by RNA-seq technology with reference transcriptome sequencing. By raw read detection and filtering, more than 50,782,597 reads were obtained from each sample (Table 2). After filtering, the number of valid bases exceeded 90.85%, the GC content reached 43.52%, and the nucleotides with Cycle Q30 values exceeded 90.85% for each sample. The accuracy of the measured data was high, which facilitated the analysis of the data at a later stage.

**Table 2.** Statistics of SA-mediated transcriptome sequencing of melon in response to salt stress.

| Sample | Raw Reads | Clean Reads | Q30 Content | GC Content (%) |
|---|---|---|---|---|
| NaCl | 50,782,597 | 46,390,802 | 90.85 | 43.52 |
| SA + NaCl | 61,157,746 | 56,421,838 | 92.10 | 43.58 |

### 3.4. Differential Gene Expression Analysis

Differential gene expression analysis was performed for all expressed genes, as shown in Figure 2A. A total of 5115 differentially expressed genes (DEGs) were obtained, including 2958 genes with up-regulated expression and 2157 genes with down-regulated expression. Gene ontology (GO) analysis of differentially expressed genes revealed that NaCl 48 h vs. SA + NaCl 48 h were significantly different in terms of biological processes, cellular components, and molecular functions. In the category of cellular components, the top three enriched GO terms were plasma membrane (GO:0031226), cytoskeleton microtubules (GO:0015630), and plant-type cell wall (GO:0009505). In the category of molecular function, the top three enriched GO terms were tubulin binding (GO:0015631), oxidoreductase activity, acting on peroxide as an acceptor (GO:0016684), and microtubule binding (GO:0015631). In the category of biological processes, the top three enriched GO terms were the cell cycle process (GO:0022402), cell division (GO:0051301), and mitotic cycle (GO:1903047).

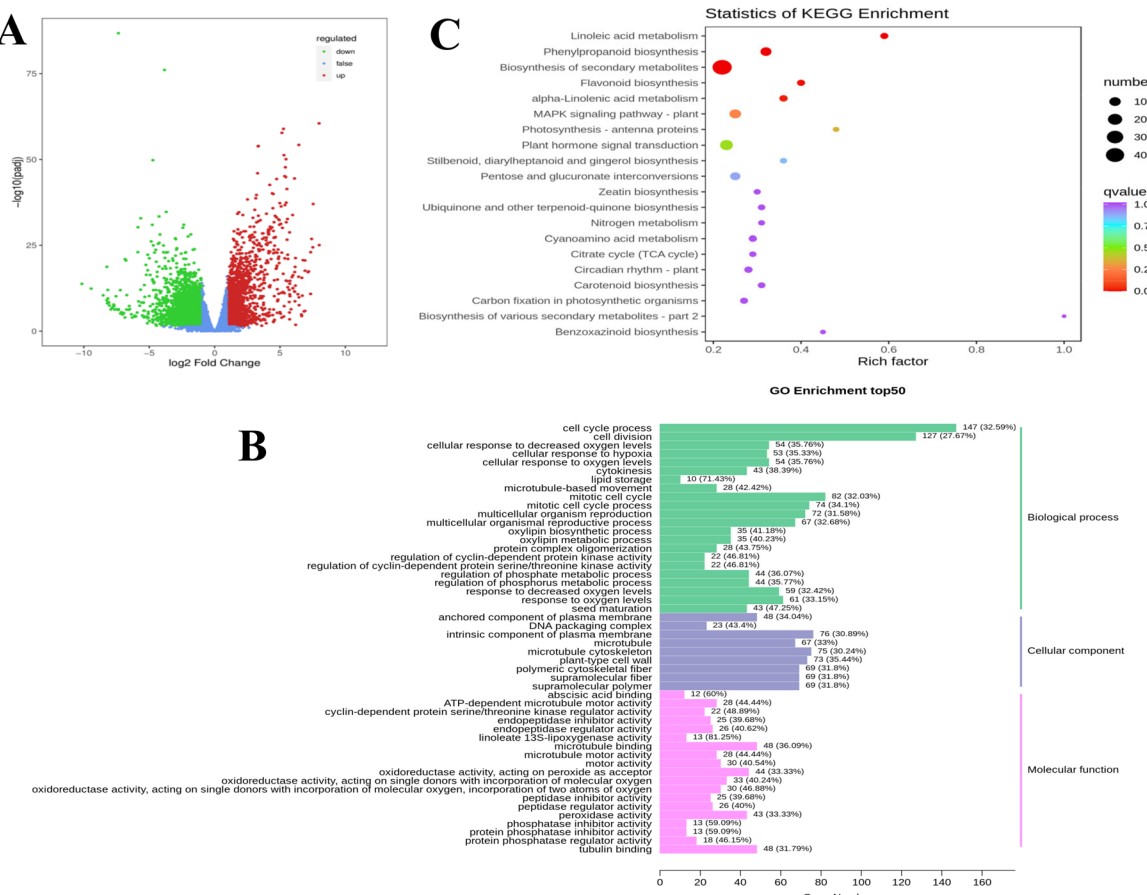

**Figure 2.** Differentially expressed genes (DEGs) analysis betweenNaCl 48 h and SA +NaCl 48 h.
(**A**) Volcano plot depicting the up-, down- and non-regulated genes between the two treatments.
(**B**) Gene ontology enrichment analysis of the DEGs. (**C**) KEGG enrichment analysis of the DEGs.

Furthermore, the Kyoto Encyclopedia of Genes and Genomes (KEGG) enrichment analysis of the DEGs showed that lipid metabolism (linoleic and α-linolenic fatty acids metabolism), signaling transduction (mitogen-activated protein kinase (MAPK) signaling pathway and plant hormone signal transduction), and biosynthesis of secondary metabolites (phenylpropanoid pathway and flavonoid biosynthesis) were the main pathways contributed by the DEGs.

### 3.5. DEGs Related to Lipid Metabolism

Seeds contain many lipoxygenases (LOXs), which have an important role in abiotic stresses. As shown in Figure 3A, Table S2, seven LOX-related genes, i.e., MELO3C014627.2, MELO3C000684.2, MELO3C014634.2, MELO3C004250.2, MELO3C014637.2, MELO3C031048.2, and MELO3C000770.2, were mapped to the pathways related to linoleic and α-linolenic fatty acid metabolism. Interestingly, all of these genes were up-regulated in SA + NaCl.

### 3.6. DEGs Related to Biosynthesis of Secondary Metabolites

The pathway of phenylpropane biosynthesis is one of plants' most important secondary metabolic pathways. As shown in Figure 3B, Table S3, ten DEGs, all being proteins related to lignin synthesis (PER), were detected, with two down-regulated (MELO3C003377.2, MELO3C028619.2) and eight up-regulated genes (MELO3C008187.2, MELO3C002391.2, MELO3C019612.2, MELO3C008188.2, MELO3C023613.2, MELO3C021914.2, MELO3C018804.2, and MELO3C020841.2) in SA + NaCl. In the flavonoid anabolic pathway, only one UDP-glycosyltransferase gene (UGT) (MELO3C019888.2) was detected, with down-regulation in SA + NaCl.

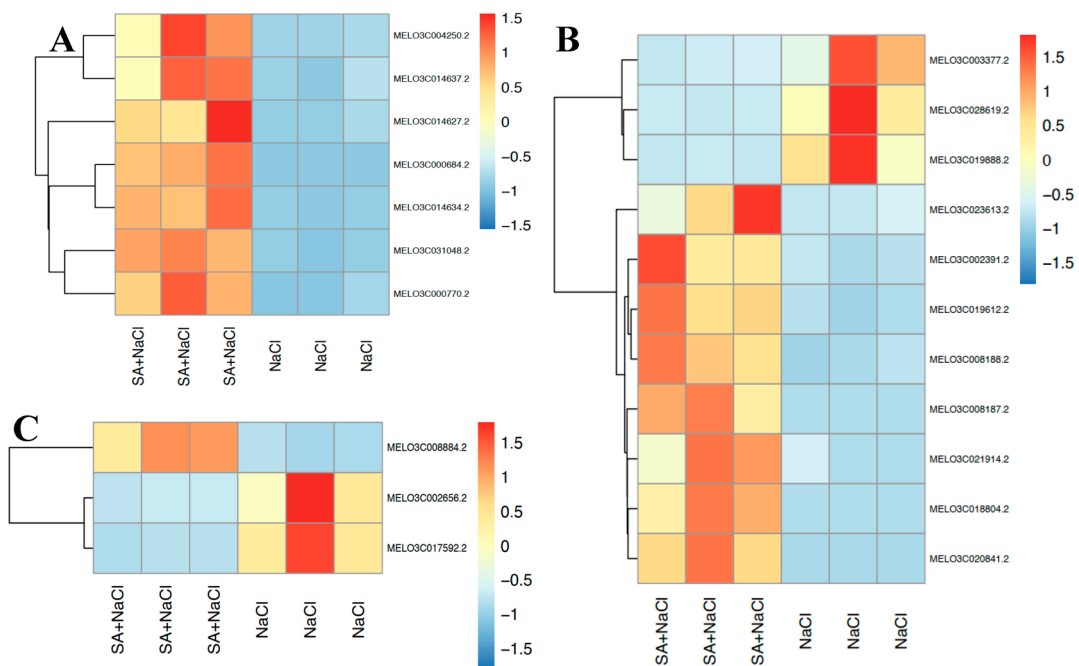

**Figure 3.** Heatmap showing the DEGs involved in different pathways in melon mediated by SA under salt stress. The colored bars represent the value (log$_2$ (fold change)) of the change in gene expression of the different treatments. Red represents up-regulated DEGs, and blue represents down-regulated DEGs. (**A**) Lipid metabolism; (**B**) biosynthesis of secondary metabolites; (**C**) signal transduction.

### 3.7. DEGs Related to Signal Transduction

In this important pathway, we found three DEGs, including one serine/threonine protein kinase SRK2J gene (MELO3C008884.2), one abscisic acid (ABA) receptor PYL4 gene (MELO3C002656.2), and one ethylene response factor EIL4 (MELO3C017592.2) gene, whereas only the SRK2J gene was up-regulated in SA + NaCl (Figure 3C, Table S4).

### 3.8. Validation of Gene Expression Using qRT-PCR

To verify the accuracy of the sequencing results, 12 randomly selected differential genes were subjected to qRT-PCR. The qRT-PCR results were strongly correlated with the RNA-seq data (R$^2$ = 0.90, Figure S2), indicating that transcriptome sequencing results were accurate and reliable.

## 4. Discussion

Soil salinity has become one of the most harmful issues to cultivatable land. Enhancing salt tolerance in melon by genetic means is an effective remedy; however, the selection of new varieties is relatively lengthy due to long breeding cycles and limited germplasm resources. In addition, applying exogenous regulatory substances is a simple and feasible method to improve salt tolerance in crops with a broad application prospect. Therefore, it is necessary to elucidate the mechanism through which SA promotes the seed germination of melon under salt stress. Although SA can induce salt resistance, which also been reported in various plants [30–34], knowledge of the molecular mechanism of melon is limited, just as potential candidate genes. Herein, aiming to explore the SA-mediated molecular response to salt stress, we performed an RNA-seq-based comparative transcriptome analysis between NaCl and SA + NaCl treatment and analyzed the transcriptomic differences between these two contrasting treatments.

SOD, POD, and CAT are important protective enzymes of the enzymatic defense system in plants, which can scavenge reactive oxygen species and have an important role in resisting abiotic stresses, such as drought, salt, cold, and heat [35]. SOD is the first line of defense against reactive oxygen radical-mediated oxidative damage; it catalyzes

the dismutation of superoxide anion radicals to $H_2O_2$ and $O_2$. POD is a heme-containing enzyme that oxidizes various substrates, such as phenolic compounds and antioxidants using $H_2O_2$ and prevents excess accumulation of $H_2O_2$ [36,37]. Topical SA application increased SOD and CAT activities throughout the treatment period; however, POD enzyme activity did not show a clear pattern, which implies that CAT and SOD may be key enzymes in the antioxidant enzyme system of melon during seed germination under salt stress regulated by SA, which is consistent with Torun's study [38].

Pro is an ideal organic osmoregulatory substance that can mitigate the damage caused by excessive water loss to cells under salt stress and guarantee the normal supply of water in plants [34]. Yet, conclusions regarding the correlation between changes in Pro content and stress resistance are not consistent across crops [39,40]. In the present study, we found that SA reduces the Pro content over time, which is similar to the effect of TDM (triadimefon) in ameliorating cold damage in cucumber seedlings [41].

The key enzyme in seeds during germination is lipoxygenase (EC 1.13.11.12, LOX), and the catalytic substrates are linolenic acid and linoleic acid. Early studies found that abiotic stresses, such as drought, can induce LOX gene expression, suggesting that the LOX pathway may mediate plant responses to abiotic stresses [42]. In *Arabidopsis thaliana*, LOX genes were highly associated with salt tolerance, and *LOX3* deletion mutants showed salt sensitivity at both germination and growth stages [43]. Furthermore, in Arabidopsis overexpression plants, *CaLOX1-OX* showed enhanced salt tolerance along with less $H_2O_2$ accumulation and significant upregulation of stress response gene expressions, such as *RD22*, *RD29A*, *RD29B*, and *P5CS*, suggesting that *CaLOX1* may be positively regulating salt stress response by regulating $H_2O_2$ accumulation and stress response gene expression [44]. In this study, seven *LOX*-related genes were mapped to the pathways related to linoleic and $\alpha$-linolenic fatty acid metabolism, and all of these genes were up-regulated in SA + NaCl. These results suggest that exogenous SA may defend against salt stress by positively regulating LOX gene expression.

Phenylpropanoid biosynthesis is a major source of several defensive secondary metabolites (e.g., lignin) in plants, and intermediates of this pathway may provide raw materials for lignin metabolism [45]. Lignin is the main component of the plant cell wall that has an important role in mechanical support and water transport [46,47], reducing damage from abiotic cellular stresses. It is also the first structure to sense and respond to environmental stresses [48]. It has been shown that the lignification of plant root cell walls increases under salt stress, which can effectively prevent ion uptake inside the cells, enhance the structural rigidity and robustness of conduction tissues and improve the salt tolerance of plants [49]. In the present study, we found that the ten DEGs, all being proteins related to PER associated with lignin synthesis, were significantly detected in this pathway, and eight genes were up-regulated in SA + NaCl. The cell wall is an important determinant of cell form and function and is the first natural barrier against stress [50]. In the face of abiotic stresses (e.g., salt stress), plant cell walls are mainly subject to structural and compositional changes [4]. In general, plant cells have dedicated systems (LRX3/4/5-RALF22/23-FER, THE1, MIK2 protein system, etc.) for monitoring the functional integrity and compositional changes of the cell wall, and some induce mechanisms to repair the damaged cell wall, including cell wall metabolism, cytoskeletal organization changes, vesicle transport, and other processes [51]. Interestingly, in the GO entry, we also found the plant cell wall (GO:0009505) and cytoskeleton microtubule (GO:0015630). The above results suggest that SA may regulate cytoskeletal dynamics and improve cell wall stability, thus contributing to cellular resilience. It has been shown that overexpression of the *CrUGT87A1* gene of *Carex rigescens* in *Arabidopsis thaliana* enhances plant salt tolerance by increasing the content of flavonoid substances [52]. In this experiment, one UDP-glycosyltransferase (UGT) gene (MELO3C019888.2) in the flavonoid synthesis metabolic pathway was highly expressed under salt stress. The reduction in gene expression levels after SA addition may be an indirect effect of SA to alleviate salt stress damage in melon rather than an active regulation.

The transcripts of one serine/threonine protein kinase SRK2J gene, one ABA receptor PYL4 gene, and one ethylene response factor EIL4 gene were significantly detected in the MAPK signaling pathway and plant hormone signal transduction. It has been shown that the MAPK pathway activates ACS genes required for ethylene biosynthesis and regulates salt tolerance [53]. The role of ABA as a central hormone in plant stress response, its synthesis, and the signal transduction pathway in salt stress response have gained increasing interest among researchers [54,55]. Under salt stress, accumulated ABA and PYL receptors in plants bind to form ABA-PYL complexes, which bind to PP2C-like phosphatases and release SnRK2 kinases (SnRK2.2, 2.3, and 2.6), thus phosphorylating numerous transcription factors downstream and in turn causing physiological responses, such as ABA-responsive gene expression, stomatal closure, and germination inhibition [56,57].

## 5. Conclusions

In this study, we used a combination of physiological and transcriptomic to investigate how SA promotes melon seed germination under salt stress. Our data indicated that SA positively affects salt tolerance by increasing the activity of SOD and CAT and decreasing the content of Pro. Further analysis revealed that SA might alleviate salt stress by initiating a series of signaling pathways under salt stress, participating in lignin biosynthesis to improve cell wall stability, and positively regulating lipoxygenase (LOX) genes. These results provide valuable information and new strategies for future salt resistance cultivation and high melon yield.

**Supplementary Materials:** The following supporting information can be downloaded at: https://www.mdpi.com/article/10.3390/horticulturae9030375/s1, Table S1: Primers for qRT-PCR validation of DEGs; Table S2: Differentially expressed genes related to lipid metabolism; Table S3: Differentially expressed genes related to biosynthesis of secondary metabolites; Table S4: Differentially expressed genes related to signal transduction; Figure S1: Response of SA-mediated phenotypes to salt stress; Figure S2: Validation of 12 DEGs by qRT-PCR.

**Author Contributions:** Formal analysis, M.Y. and J.M.; methodology, G.H. and Q.H.; validation, T.W., T.X. and H.W.; writing—original draft, M.Y. All authors have read and agreed to the published version of the manuscript.

**Funding:** This work was supported by the Opening Project of Xinjiang Key Laboratory of Crop Biotechnology (XJYS0302-2022-3), the Specific Research Fund of The Innovation Platform for Academicians of Hainan Province (No. SQ2021PTZ0063), and the China Agriculture Research System of MOF and MARA (CARS-25).

**Data Availability Statement:** All datasets obtained for this study are included in the manuscript/Supplementary Materials.

**Conflicts of Interest:** The authors declare no conflict of interest.

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
