# Peer review of "Transcriptomic Analysis of Salicylic Acid Promoting Seed Germination of Melon under Salt Stress"

_horticulturae, doi:10.3390/horticulturae9030375_

Round 1

Reviewer 1 Report

As the study is only about effect of SA on NaCl-treated seeds, how do we know that NaCl had any negative effect on seeds, as there is no non-saline control? It seems to be OK to perform transcriptomic analysis only with seed material from these two treatments, but proper germination test needs to be performed with 4 treatments: control, SA, NaCl, NaCl + SA, with necessary number of seeds and statistical evaluation of suitable germination characteristics. It is desirable that also biochemical analyses are performed on these 4 treatments, for better comprehensibility.

Introduction

Seems to be too short and descriptively general. It needs to be become clear why this particular study was necessary. There are similar studies performed with plant seeds at germination stage. Therefore, it is necessary to describe clearly what particular problems this study will solve. It is not enough to indicate "to provide a theoretical basis for the application of SA", as this seems to be not a practically-oriented study.

The term "functionalomics" seems to be pure slang. Why not use "transcriptomics" instead?

Materials and methods

Clearly indicate the number of replicates and the number of seeds per replicate used for each type of analysis. 

In 2.2., necessary details are missing to evaluate performed actions or to repeat the experiment.

Statistical analysis needs to be performed with germination and biochemical data, therefore, this needs to be included also in M&M.

Results

Include results from seed germination tests with control, SA, NaCl and SA + NaCl-treated seeds, provide statistical evaluation.

Provide statistical evaluation of results in Figure a, b, c, d. At present, description of these results in the text has nothing to do with real changes without the analysis of significance of differences and can be due to stochastic variation resulting from low number of seeds/samples during analysis (which was not specified).

Line 131, what is meant by "six samples", 3 from NaCl-treated and 3 from NaCl + SA-treated seeds? 

In 3.3., indicate that code names mean genes whose expression changes. Try to use meaningful description of these gene functions, instead of meaningless expressions as "intrinsic components of membrane plasma" (it is "plasma membrane"!). 

In Figure 2b, gene annotations need to be readable. 

In 3.4., do not include unsupported information related to discussion about role of seed LOX in abiotic stress. 

Line 168, not proteins were enriched, but rather transcripts of genes coding for these proteins.

There are no references in the text to Table S2, Table S3, Table S4, but it is necessary

Discussion

It is necessary to start with a general overview what hypotheses were tested in this study.

There are extremely many factual and grammatical inaccuracies throughout Discussion.

Try to use language expressions corresponding to reality, namely, effects of SA on gene expression under salinity, instead of using slang expressions implying that both genes and proteins were enriched, and even cell components were enriched.

Lines 194–195, SOD does not convert O2 to H2O2 and O2, it catalyzes dismutation of superoxide anion radical to H2O2 and O2. 

Line 209, SA cannot convert Pro into amino acids, it can stimulate enzyme activity that does it. Proline itself is an amino acid!

Line 242, plant cells do not induce cell wall metabolism, abiotic stresses result in changes in cell wall metabolism.

List of references needs to be formatted according to the journal standards.

Reviewer 2 Report

I kindly ask you to see my observations and recommendations that have been done on the manuscript and consider them.

Round 2

Reviewer 1 Report

Thank you for performed additional experiments and rigorous corrections of the manuscript

Author Response

Thank you for your help.